# The architecture and stabilisation of flagellotropic tailed bacteriophages

Joshua M. Hardy [1,6], Rhys A. Dunstan [2,6], Rhys Grinter [2], Matthew J. Belousoff[2], Jiawei Wang[2], Derek Pickard[3,4], Hariprasad Venugopal[5], Gordon Dougan[3,4], Trevor Lithgow [2✉] & Fasséli Coulibaly [1✉]

Flagellotropic bacteriophages engage flagella to reach the bacterial surface as an effective means to increase the capture radius for predation. Structural details of these viruses are of great interest given the substantial drag forces and torques they face when moving down the spinning flagellum. We show that the main capsid and auxiliary proteins form two nested chainmails that ensure the integrity of the bacteriophage head. Core stabilising structures are conserved in herpesviruses suggesting their ancestral origin. The structure of the tail also reveals a robust yet pliable assembly. Hexameric rings of the tail-tube protein are braced by the N-terminus and a β-hairpin loop, and interconnected along the tail by the splayed β-hairpins. By contrast, we show that the β-hairpin has an inhibitory role in the tail-tube precursor, preventing uncontrolled self-assembly. Dyads of acidic residues inside the tail-tube present regularly-spaced motifs well suited to DNA translocation into bacteria through the tail.

[1] Infection & Immunity Program, Biomedicine Discovery Institute & Department of Biochemistry and Molecular Biology, Monash University, Clayton, VIC, Australia. [2] Infection & Immunity Program, Biomedicine Discovery Institute & Department of Microbiology, Monash University, Clayton, VIC, Australia. [3] Wellcome Trust Sanger Institute, Hinxton, Cambridge CB10 1SA, UK. [4] Department of Medicine, University of Cambridge, Addenbrooke's Hospital, Hills Road, Cambridge, UK. [5] Ramaciotti Centre for Cryo-Electron Microscopy, Monash University, Clayton, VIC, Australia. [6]These authors contributed equally: Joshua M. Hardy, Rhys A. Dunstan. ✉email: trevor.lithgow@monash.edu; fasseli.coulibaly@monash.edu

P hages represent the most abundant organisms on the planet[1]. Of the known phages, most have double-stranded DNA genomes and belong to the *Caudovirales* order[2,3], characterised by an icosahedral head and a helical tail. The tail can be very short (*Podoviridae*, e.g. T7), long and contractile (*Myoviridae*, e.g. T4) or long and non-contractile (*Siphoviridae*, e.g. λ and χ)[2,3]. The head and tail structures are assembled independently: the phage DNA is pumped into the head to reach internal pressures of around 20–60 atm[4–6]; only after genome packaging and capsid maturation are complete is the fully assembled tail component joined to form the infectious particle.

The head and the tail components of phages have to withstand considerable mechanical stress in the extracellular phase of the viral cycle[7], perhaps none more so than flagellotropic phages. These phages, such as the archetypal χ phage[8,9], recognise and engage bacterial flagella before being spun at extraordinary velocity (several microns per second) and shear force towards the bacterial cell surface[10] (Supplementary Fig. 1). Integrity of the tail is essential to the viability of the phage, being required for the injection of the genome into the bacterial host[11]. In contractile tails of myoviruses, an outer sheath braces the entire central tail-tube in the pre-contracted state, presumably adding stability to the assembly. Given that they have no outer tube structure and are up to twice as long, it is of interest to understand how tail-tube integrity is maintained in flexible tails typical of the *Siphoviridae*. The best available model of such a tail is derived from a pseudo-atomic model for phage T5[12]. How the ~10 μm (~60 kbp) genome efficiently transits through this long narrow tunnel is also not fully understood. Calculations suggest that frictional forces within the tail should greatly hinder movement[13] and theoretical constraints from our current knowledge can account for only 15% of the length of the genome leaving the phage[5]. The lack of high-resolution structural information on the assembled non-contractile tails has stymied a detailed understanding of its assembly from monomeric tail-tube proteins and its role in facilitating DNA egress.

By contrast, the heads of numerous tailed bacteriophages have been structurally characterised revealing an intricate chainmail-like organisation of the major capsid protein[14–16]. The HK97-like capsids assemble into a bona fide chainmail formed by a covalent crosslinking of the major capsid protein[17,18]. Extending the strict definition of protein chainmail to include chains that are non-covalently interlocked, three other classes of chainmail-like organisations have been defined[19]: (1) the T4 and P22-like capsids that are stabilised by an inserted domain in the minimal HK97 fold[20–22]; (2) the BPP1-like capsids that use a major capsid protein in which structural elements are permuted compared to the canonical HK97 topology[23]; and (3) the λ/χ-like capsids. In the absence of a covalent chainmail, additional proteins called decoration, auxiliary or cementing proteins may add to the stability of the particle. There are no high-resolution structures of the model phage λ or the archetypal flagellotropic phage χ belonging to the third class.

Here we have determined by cryo-electron microscopy (cryo-EM) structures of the head and tail components of YSD1, a phage infecting *Salmonella* Typhi. Comparison to the genome sequence of the χ-phage indicated that YSD1 and χ-phage share structural proteins with identical amino acid sequences[24]. At a resolution of 3.8 Å, the structure of the YSD1 head provides a model for the λ/χ-like capsids presenting unique similarities with herpesvirus virions. These capsids share an external non-covalent chainmail of the auxiliary protein, which adds to the stability of the particle. The 3.5 Å-resolution structure of the YSD1 tail reveals a robust but pliable assembly based on hexameric rings strung around a central spine of the tape measure protein. An extended β-hairpin plays a dual role in assembly, mediating most of the inter-ring contacts in the assembled tail but preventing uncontrolled self-assembly of the tail-tube precursor.

## Results

**Overall structure of the YSD1 virion**. We investigated the structure of the recently characterised phage YSD1[24] as an example of flagellotropic χ-like phages and more generally phages with long, non-contractile tails. Samples of YSD1 imaged by EM showed characteristic features of siphoviruses: an icosahedral capsid and a flexible tail tube of ~220 nm in length (Fig. 1a; Supplementary Fig. 1a). Attached to the tail tube of YSD1 is a long, flexible tail fibre[24].

Cryo-EM analysis of YSD1 allowed structure determination of the head and tail components (Fig. 1a–c) at resolutions of 3.8 Å and 3.5 Å, respectively, and the generation of de novo models for the major capsid (YSD1_17), auxiliary protein (YSD1_16) and tail-tube protein (YSD1_22) (Supplementary Table 1). YSD1 has an icosahedral head organised with a *T* = 7 *laevo* symmetry (Supplementary Fig. 2). The mature head contains 415 major capsid proteins arranged in hexon and penton capsomers, forming a shell that is remarkably thin for its ~650 Å diameter. This shell is further decorated by trimers of the auxiliary protein located at each threefold or pseudo threefold position (Figs. 1 and 2, Supplementary Fig. 2a, b).

**The E-loop and N-terminal arm interlock the capsid capsomers.** While the general organisation is similar to the other classes of HK97-like capsids, features in the major capsid protein fold and a unique role of the auxiliary protein in λ/χ-like capsids reveal distinctive means to stabilise the capsid. To investigate the assembly of the major capsid component, we determined the structures of the major capsid protein both within the assembled head by cryo-EM and in its soluble form by X-ray crystallography of the

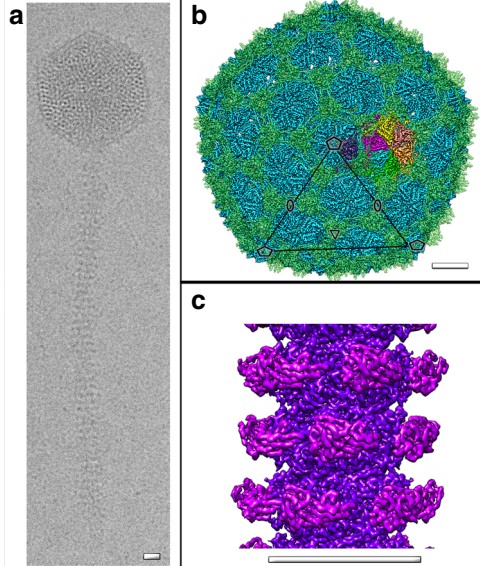

**Fig. 1 Structural architecture of YSD1. a** Vitrification conditions were optimised to trap the flexible *Siphoviridae* tail tubes as relatively straight segments. Representative cryo-EM image of a YSD1 virion (five imaging sessions). Cryo-EM reconstruction of the YSD1 head (**b**) and tail tube (**c**). Major capsid protein (YSD1_17) is shown in blue, auxiliary protein (YSD1_16) is shown in green and one icosahedral asymmetric unit is highlighted in colour (cf. Supplementary Fig. 2 for details). Icosahedral axes are indicated by ellipses, triangles and pentagons for twofold, threefold and fivefold symmetry axes, respectively. Scale bars = 10 nm.

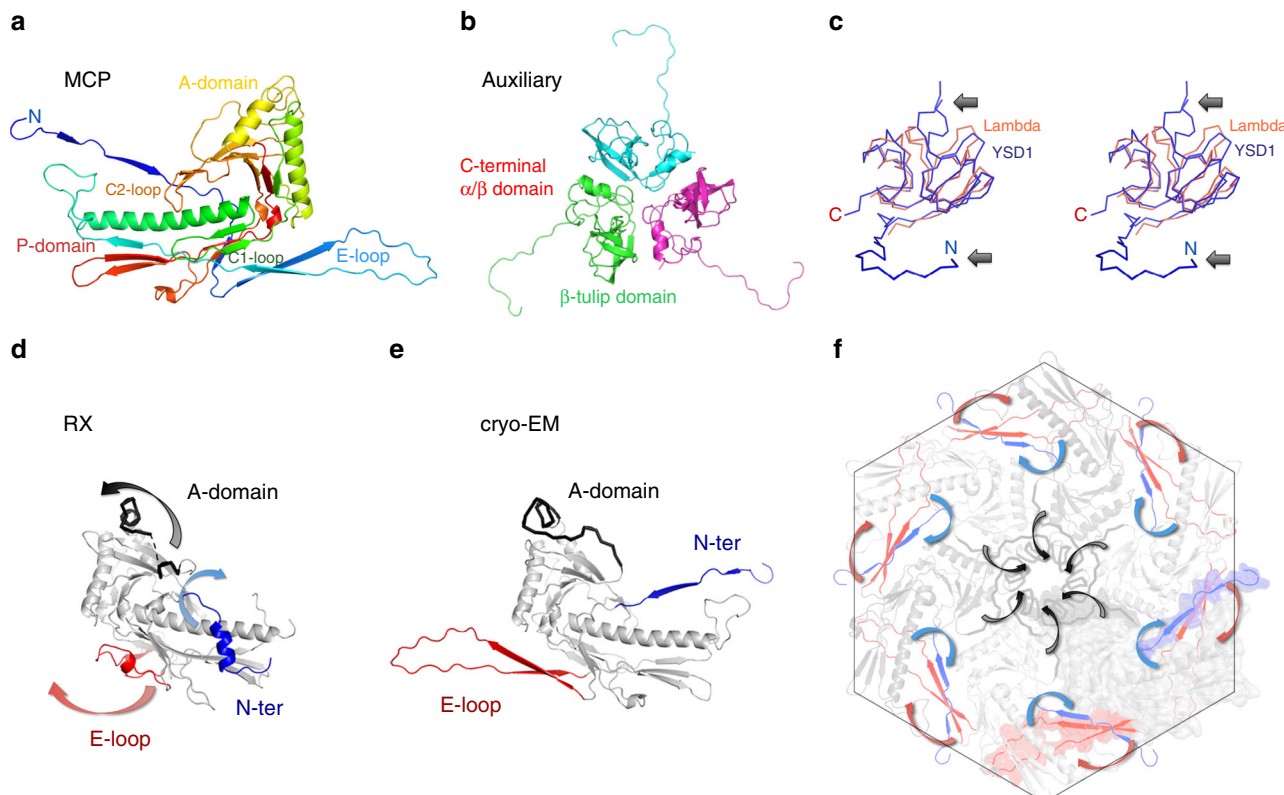

**Fig. 2 The major capsid and auxiliary proteins. a** Cartoon representation of the major capsid protein with blue–red gradient from N- to C-terminus. **b** The auxiliary protein is a homotrimer composed of two domains with a β-tulip fold and an N-terminal hook. **c** Wall-eyed stereo image of the homologous λ cementing protein, gpD (PDB ID 1C5E, orange) aligned with YSD1_16 (blue). Arrows highlight the two main differences in the N-terminal region and β-tulip domain. **d**, **e** Cartoon representations of the major capsid protein structures (grey) derived from a crystal structure of the monomeric protein (RX, **d**) and the cryo-EM reconstruction (**e**). Conformational switches of the E-loop, N-ter and A-domain regions are highlighted by arrows. **f** Hexon capsomer in the cryo-EM structure of the icosahedral head.

recombinant protein (resolution: 2.6 Å, Supplementary Table 2). As expected, in both the soluble and assembled forms, the major capsid protein has two domains typical of the HK97 fold[18] (Supplementary Tables 3 and 4): the peripheral P-domain forming the outer rim of pentons and hexons in the virion, and the wedge-shaped, axial A-domain that packs tightly at the centre of the capsomers (Fig. 2a; Supplementary Fig. 3). Comparison of the two structures revealed major rearrangements correlating with the transition from a soluble, monomeric form to an assembled capsid. These rearrangements are unlikely to be caused by crystal packing since the same compact conformations are seen for both of the independent molecules in the asymmetric unit of the crystal despite completely different environments. Taking the compact crystal structure as a reference, these rearrangements can be described as the stretch of the N-terminal arm and E-loop in opposite directions with regard to the central P-domain (Fig. 2). In these conformations, the two structural elements form an inter-molecular beta-sheet with their counterparts of the neigboring subunits within a capsomer, which results in a circular strand exchange. In addition to interlocking the capsomer, these inter-actions allow for quasi-equivalent interactions modulated by varying bending angles of the N-terminal arm and E-loop. These quasi-equivalent interactions observed in the asymmetric unit of the capsid (Supplementary Fig. 2) ultimately allow the formation of pentamers and hexamers with different curvatures, which are required to assemble the large T7 icosahedral shell.

**Two nested sets of chainmail stabilise the phage head.** Inter-actions involving the extended E-loops at the threefold axis

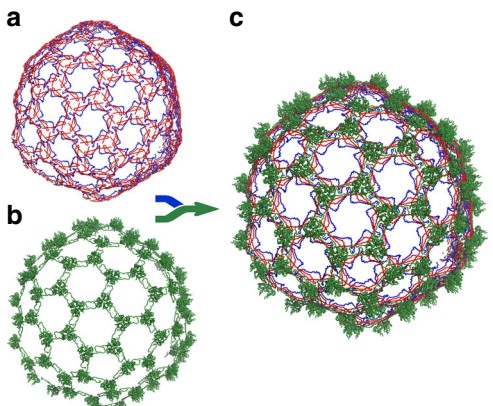

**Fig. 3 The capsid proteins form two nested chainmails. a** Representation of the non-covalent chainmail formed by the N-terminal arms and E-loops of the major capsid protein. The representation is the same as Fig. 2 for N-ter and the E-loops. The rest of the capsid protein has been omitted. **b** Ribbon representation of the outer chainmail of the auxiliary protein. **c** Representation of both chainmail structures.

further connect each capsomer into a non-covalent chainmail represented in Fig. 3a. This chainmail contributes to the stability of the capsid through two components not found in the other classes of phage major capsid proteins (Supplementary Fig. 4). First, a clamp formed by two loops, C1 and C2 immobilises the E-loop around the threefold axes (Fig. 4a). Second, trimers of the auxiliary protein dock at the threefold axes over the clamp and

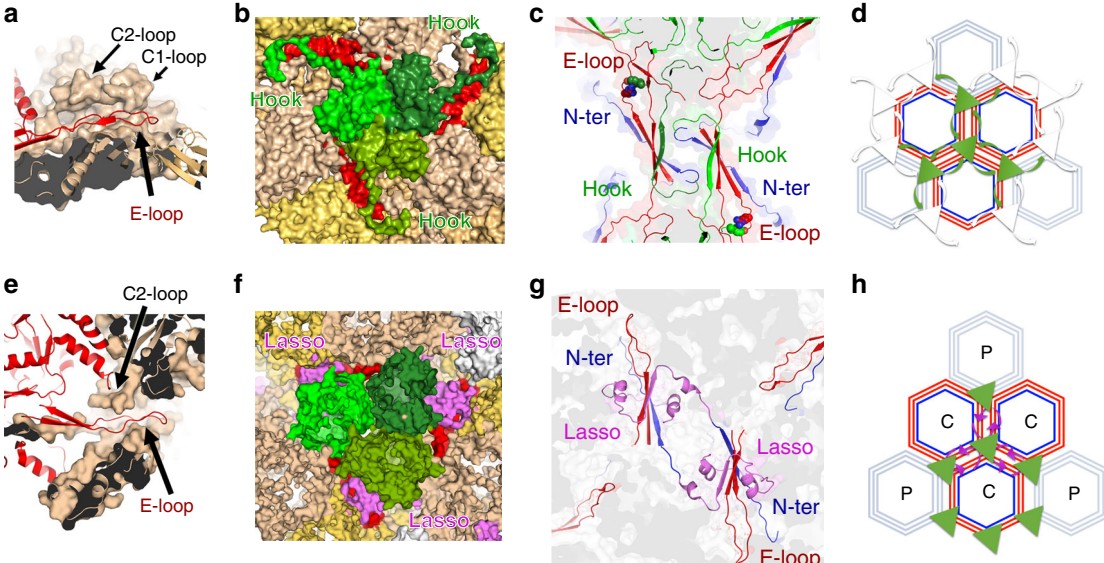

**Fig. 4 λ/χ-like phages evolved capsid stabilisation strategies that have been maintained in herpesviruses.** Surface representations of the major capsid protein shells in the YSD1 (**a**) and human cytomegalovirus (HCMV, PDB ID 5VKU) (**e**) viruses with selected molecules shown as a cartoon only (red) to highlight clamping of the E-loop. Surface representation of a pseudo threefold axis with the MCPs in shades of brown and auxiliary proteins coloured in green (**b**, YSD1_16 homotrimer; **f** heterotrimer of HCMV triplex proteins). The YSD1 E-loop and HCMV lasso are coloured in red and violet respectively. **c**, **g** Close-up view of the inter-molecular β-sheets that staple two hexons. The N-terminal arm and E-loop of the major capsid protein are coloured in blue and red as in Fig. 2. The 4th β-strand is shown in green and corresponds to the N-terminal hook of YSD1_16 (**c**) or the N-terminal lasso domain of the MCP from the neighbouring hexon in HCMV (**g**). Cartoon representations of chainmail networks in YSD1 (**d**) and HCMV (**h**). Hexagons represent hexons of the MCP with the N-ter, E-loop and hook/lasso shown with the same colour scheme as other panels. For HCMV, the peripentonal and central hexons are noted with P and C, respectively. Triangles represent the trimeric auxiliary protein.

E-loop locking in place this critical capsomer interface (Fig. 4b–d). The auxiliary protein has a tulip fold and trimeric organisation similar to the gpD protein of the λ phage with a root mean square deviation (RMSD) between Cα of 88 equivalent residues of 1.7 Å (Fig. 2b, c; Supplementary Table 5). In both YSD1 and λ, the N-terminus of the auxiliary protein staples neighbouring capsomers by interactions around the twofold axis augmenting the major capsid β-sheet of the E-loop/N-terminal arm (Figs. 3c, 4c, d; Supplementary Fig. 5). However, in the flagellotropic χ-like phages, an additional hook-like structure formed by an N-terminal extension connects trimers of the auxiliary proteins (Figs. 2c, 3b; Supplementary Fig. 5). This interaction mediates the formation of an external protein network that braces the inner capsid shell (Fig. 2b, c; Supplementary Fig. 5d). As in the thermophilic phage P23–45[25], this outer chainmail is likely to contribute to environmental stability, important for flagellotropic phages when subjected to high shear forces on a flagellum, or the high temperatures faced by thermophilic phages.

Although there is no structure of the λ phage major capsid protein, comparison with the 6.8 Å-resolution cryo-EM electron density map of the λ head[26] with the YSD1 structure supports similar stabilising roles of the clamp (Supplementary Fig. 3) and the auxiliary protein at the threefold vertices (Supplementary Fig. 5d). Thus, the structure of YSD1 head provides further support to the close relationship established between the χ-like and λ phages and establish the clamp and tulip auxiliary proteins as distinctive features for this class of viruses.

**χ/λ phages share stabilisation strategies with herpesviruses.** Past work on the structure and assembly mechanisms of herpesvirus capsids have led to a proposition that herpesviruses are derived from a phage ancestor[27,28]. This structural conservation is striking given the lack of sequence similarity between the major capsid proteins of phage and herpesvirus, and the evolutionary distance between these viruses[29–31]. In YSD1, features specific to λ/χ-like phages further support this structure-based homology. Comparison of core components of the capsid shows that alpha, beta and gamma-herpesviruses share the three main stabilising features of YSD1, namely the C2 clamp (Fig. 4a, e), threefold "tulip" auxiliary proteins (Fig. 4b, f) and inter-capsomer stapling at the twofold axis through an inter-molecular β-sheet augmentation (Figs. 4c, d and 3e, f). Conservation of their function and topology (Supplementary Fig. 4) suggests that the clamp and auxiliary protein of λ/χ-like phages are homologous to their herpesvirus counterparts. By contrast, there are lineage-specific features, which may have evolved to impart further stability, namely the "hook" of the YSD1 auxiliary protein and the "lasso" structure in herpesvirus major capsid protein. The conserved position of these two different structural reinforcements highlights the importance of stabilising the seams between adjacent capsomers, which may be particularly critical in such large but thin capsids.

**Structure of the tail tube.** Contrary to the capsid, the high-resolution structure of non-contractile tails has remained elusive, presumably because of their inherent flexibility. Our structure of the YSD1 tail reveals that the helical tube is built from hexameric rings of the three-domain tail-tube protein. The rings are stacked with an axial rise of 41.2 Å and a twist of 19.7° to form a right-handed helix (Fig. 5a; Supplementary Fig. 6). Clear density in the cryo-EM helical reconstruction (Fig. 5a, b) allowed structure determination of the first two domains (Fig. 5c). Only weak density is present for the third, C-terminal domain, indicating that it does not strictly follow the helical symmetry of the tail tube.

The main body of the tail tube is formed by domain 1, a sandwich composed of two four-stranded β-sheets with a long

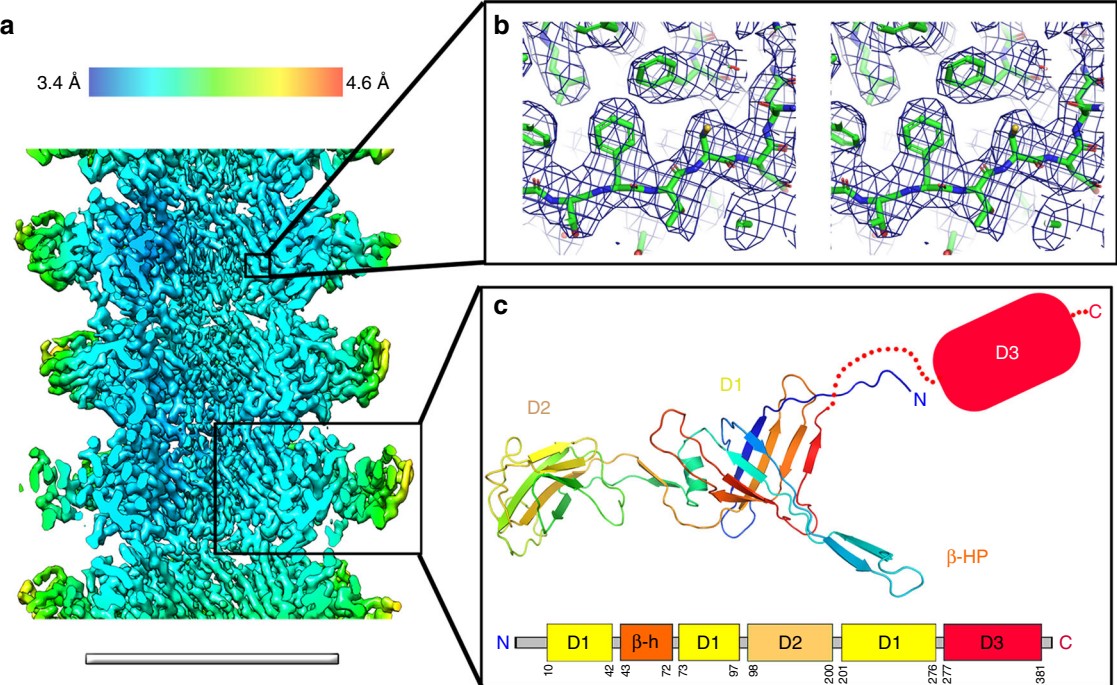

**Fig. 5 Cryo-EM structure of the YSD1 tail. a** Surface representation of the cryo-EM helical reconstruction of the tail tube. The colour gradient represents the local resolution of the map. Scale bar = 10 nm. **b** Stereo diagram showing the electron density map and de novo model of the tail protein. **c** The tail tube protein YSD1_22 is composed of three structural domains and an extended β-hairpin loop (β-HP). Domains 1 and 2 were modelled in the cryo-EM density, while only weak density was present for domain 3 (D1-3; cf. Supplementary Fig. 8).

β-stranded loop extending strands β3 and β4 (β-hairpin). The fold of domain 1 is similar to tube-forming proteins found in contractile and non-contractile phage tails, as well as R-pyocins and bacterial type 6 secretion systems (T6SS) (Supplementary Fig. 7; Supplementary Table 6). Assembly of the tail tube involves the formation of a continuous antiparallel β-barrel connecting the largest β-sheets of six subunits on the internal face of the tail tube (Fig. 6a, b). Contrary to the tail of the T4 and T5 phages, where structural analysis showed a smooth round conduit[12,32], the inner cross-section of the YSD1 tail has a marked hexagonal shape (Supplementary Fig. 7d). This atypical shape is caused by residues $Pro_{240}$-$Asp_{241}$-$Gly_{242}$ that create a stagger in the otherwise continuous β-barrel. This feature allows the accommodation of helix H1 wedged at the interface between two adjacent subunits (Fig. 6a). H1 participates to most of the intra-hexameric contacts forming a predominantly hydrophobic interface. Surface analysis predicts that this interface imparts considerable stability to the ring, burying 1668 Å$^2$ in each intra-ring surface, which accounts for a solvation energy of $-16.6\,kcal\,mol^{-1}$ (Supplementary Table 7). This interface is only moderately conserved in sequence with much of the conserved surface residues clustering in the N-terminus, the β-hairpin and the inner surface of the ring (Fig. 6c, d).

**χ-specific features of the tail tube.** The external surface of the YSD1 tail tube is rugged compared to the smooth phage T4 tail and those of pyocins and T6SS, which all have an external sheath. This aspect is due to an additional structural feature, domain 2 inserted between strands 4 and 5 of the inner and outer sheets, respectively. Domain 2 is a β-sandwich in which two 3-stranded antiparallel β-sheets adopt a DE-variant Ig-like fold, such that an extended loop between strands 3 and 4 locks in as the third layer of the sandwich (Fig. 5c). Sequence analysis indicates that this is a distinctive feature of χ phage and its relatives. Weak structural similarity suggests a possible link to adhesion proteins

and phage/T6SS baseplate proteins (Supplementary Table 8). In the assembled tail, domain 2 stabilises the hexameric ring by wrapping around adjacent subunits and locking them into place like a latch (Fig. 6b). This interaction buries a significant surface area per subunit of 950 Å$^2$. By contrast with the central tube β-barrel formed by domain 1, this interface is highly polar with nine salt bridges (Fig. 6b). The orientation of domain 2 is maintained by interactions between the two β-strands of the tail-tube protein neck and the outer β-sheet of domain 1 of a neighbouring sub-unit, resulting in an inter-molecular β-sheet. This orientation does not allow domain 2 to form interactions between successive hexameric rings along the axis of the tail. Thus, domain 2 girds the main tube providing additional stabilisation of the hexameric ring in the absence of the external sheath found in contractile tails (Figs. 6b and 8b, f).

Domain 3 of the tail-tube protein is more flexible than the main tube and there is no evidence that it contributes to stability of the phage tail (Supplementary Fig. 8). Sequence analysis showed domain 3 is a Bacterial Ig-like domain 1 (Big-1), which is related to IgSF domains, a highly adaptable fold that can function for protein–protein[33,34] or protein–carbohydrate interactions[35]. Either type of interaction could assist phage binding to the surface of its host (e.g., to cell surface lipopolysaccharides or a secondary protein receptor)[35,36]. Based on CLuster ANalysis of Sequences (CLANS) of the tail-tube protein sequences from the 20 *Siphoviridae* clusters[9] (Supplementary Table 9), χ-like phages form a unique cluster characterised by the Big-1 domain. The majority of the other tail-tube protein clusters, including phages T5 (Lytic4 cluster) and λ (Temp1 cluster), are interconnected due to the presence of Big-2 domains or related IgSF-like domains (Supplementary Fig. 7b). In a CLANS analysis, using sequences from which the non-structural domain 3 (Big-1/Big-2) was deleted, connections between clusters are lost indicating that the ancient tail-tube-forming domains have diverged beyond sequence recognition. Overall, the data are consistent with a more

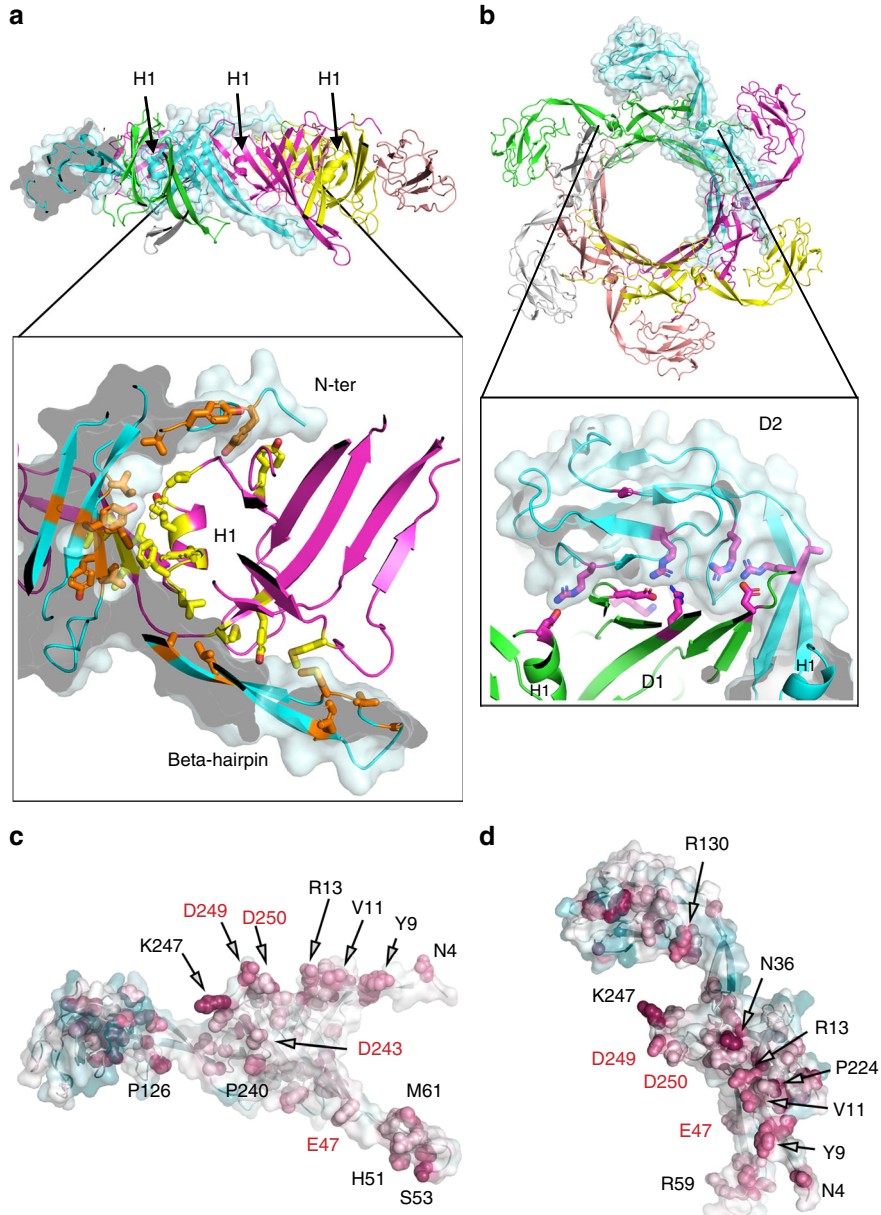

**Fig. 6 The tail tube is stabilised by extensive contacts within the hexameric ring of YSD1_22. a** The hexameric ring of the tail-tube protein is shown from the side with H1 helices indicated. In the inset, hydrophobic and aromatic residues within 4 Å of the neighbouring subunit are shown as sticks and their carbon atoms are coloured in yellow and orange. For clarity, only two subunits are represented in the zoomed view. **b** Axial view of the hexameric ring. The inset shows a zoomed view of the interface between D1 and D2. Charged residues within 5 Å of the neighbouring subunit are shown as sticks and their carbon atoms are coloured in magenta. **c**, **d** Same views as the cyan molecule in **a** and **b**. Low-to-high sequence conservation as calculated by the ConSurf server is mapped onto the molecular surface with a cyan–white–red gradient. Side chains of the most conserved residues are shown as spheres. Acidic residues are labelled in red.

recent acquisition of the Big-1 domain in tails of *Siphoviridae* that have adapted to target bacterial flagella.

**The internal surface of the tail facilitates DNA ejection.** The tail tube forms a 250 nm long, 7 nm wide corridor for the ~10,000 nm-long genome. Theoretical hydrodynamic considerations for the movement of a linear DNA molecule along such a narrow tube had to invoke a ratcheting mechanism to assist in the final stages of DNA egress, as the predictions state it would otherwise lose substantial velocity to friction and fail to move against the internal osmotic pressure of the bacterial cell[13]. The resolution of the YSD1 tail-tube structure is sufficient to map the electrostatic

surface. In contrast to a previous model of the λ tail tube[37] based on the nuclear magnetic resonance (NMR) structure of the major tail protein[38], this analysis reveals a strongly negatively charged interior (Fig. 7a) as expected for a viral DNA conduit[39]. The placement of negatively charged aspartate and glutamate residues in the assembled tail tube appears as a right-handed helical track through the tail tube (Fig. 7b, c). Most of these residues are present as dyads and their spacing is similar to the dimensions of the minor and major grooves of B-form DNA (Fig. 7c–f; Supplementary Movie 1), a feature that could potentially engage and ratchet the movement of the DNA through the helical twist inherent in the polynucleotide. We can provide no experimental test of these observations, but note that similarly spaced dyads of

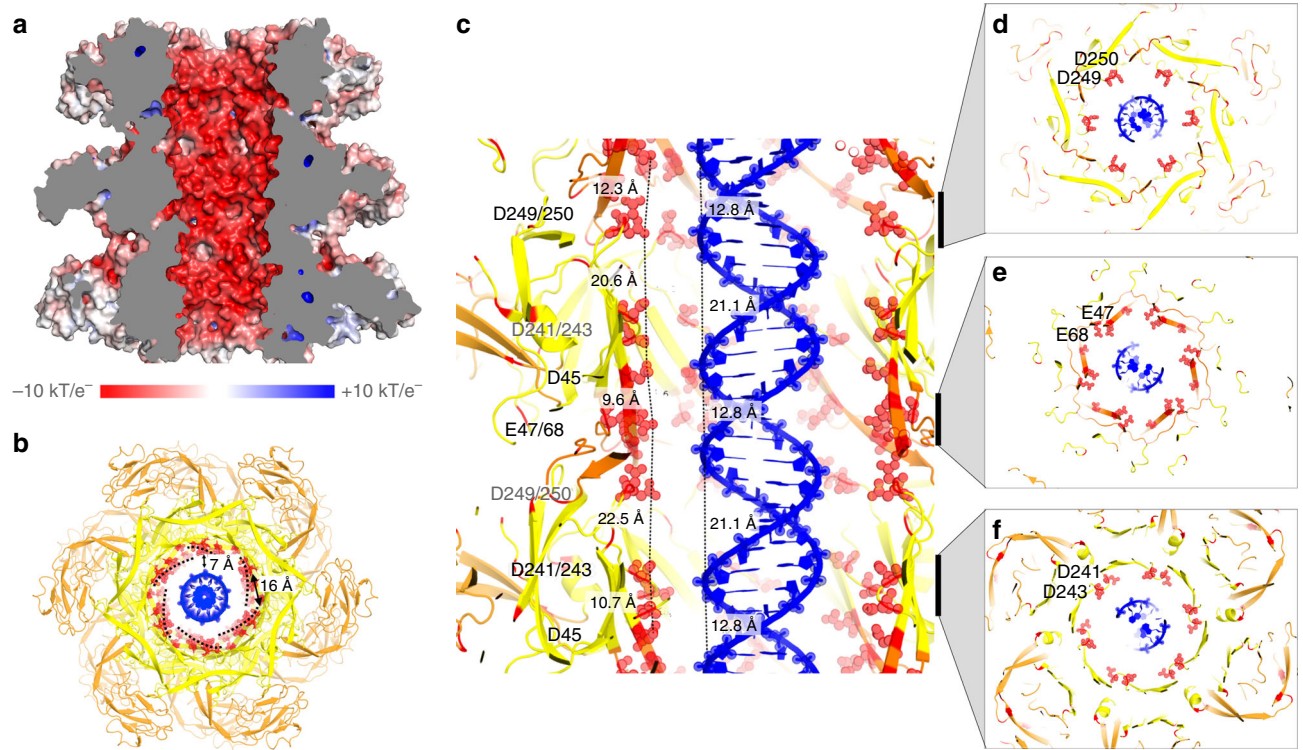

**Fig. 7 Acidic motifs in the tail tube in the context of translocating DNA.** Cut-away views of the tail tube with a double-stranded DNA segment represented at scale within the transit corridor. **a** The surface electrostatic potential is represented on the molecular surface as a red–blue gradient from electronegative to electropositive. **b–f** DNA is shown in blue and the tail tube domains 1 and 2 in yellow and orange, respectively. Acidic residues lining the inner surface of the tail tube are shown as red spheres and labelled. In **d–f** longitudinal sections of the tail viewed from the phage head highlight the presence of dyads of acidic residues. Supplementary Movie 1 and Supplementary Fig. 9 provide more details.

acidic residues are present in the T5 structure despite its different organisation based on trimeric rings with a quasi sixfold symmetry rather than hexameric rings (Supplementary Fig. 9a–f). Mapping of evolutionary conservation onto the tail-tube protein structure identified residues E47, D243, D249, D250 as being highly conserved (Fig. 6c, d). Identification of these motifs in distantly related phages for which sequences cannot be aligned with YSD1_22 will require structure-based analysis to confirm their locations in the assembled tails. However, candidate motifs are present in all the tail-tube protein sequences analysed here (Supplementary Fig. 9g; Table 9) suggesting that this structural feature is general to all *Siphoviridae*.

**Assembly of a flexible yet robust tail tube.** The apparent stability of the rings in assembled tails described above contrasts with a high solubility of the monomeric tail tube protein when expressed recombinantly. In the bacterial cytoplasm, the biogenesis of phage tail tubes requires an initiator, which has been proposed to be components of the tail tip[40,41] and/or one of the molecular chaperones coating the tape measure protein located in the lumen of the tail tube[42,43]. We used small angle X-ray scattering (SAXS) to investigate the changes required for conversion of tail-tube protein monomers into the growing YSD1 tail tube during phage morphogenesis. The tail-tube protein was expressed recombinantly without any of the putative initiators in the bacterial cytoplasm, where it formed a soluble, monomeric species. SAXS measurements on the purified protein yielded a de novo low-resolution model (Supplementary Figs. 10, 11a and Table 10). Comparison of this model with the structure of the tail-tube protein in the assembled tail positions the C-terminal domain 3 away from the tube, which is compatible with the additional density observed in cryo-EM at low thresholds (Supplementary

Fig. 8). Regions that are not accounted for by the SAXS envelope are the extended β-hairpin, the N-terminal arm and loops on the head-proximal side of the hexameric ring (Supplementary Fig. 11a). These differences with the cryo-EM structure could be due to the lack of resolution of the SAXS envelope, more compact conformations or flexibility in solution. Supporting this later hypothesis, normal mode analysis (NMA) of the structure shows that models which account best for the SAXS data have similar orientations of domains D1, D2 and D3 but differ in the conformations of the N-terminus and β-hairpin (Supplementary Fig. 11b). Conformational flexibility of these regions has also been described in the NMR structure of monomeric gpV, the major tail protein of phage λ[38] (Supplementary Fig. 11c).

To test whether these regions are involved in the transition between a highly soluble monomeric tail-tube and assembled tails, tail tube proteins lacking the N-terminus (residues 1–8, delta-N) and/or the β-hairpin (residues 45–69, delta-HP) were produced. When analysed by size-exclusion chromatography (SEC; Fig. 8a), the wild-type protein is mostly monomeric with a sub-population of a higher oligomeric species. The delta-N protein is entirely monomeric with a reduced ability to self-interact. On the contrary, the delta-HP protein readily self-assembles to form oligomers and no monomeric tail tube protein is observed. Instead, the predominant species has a molecular mass compatible with a hexamer and a cyclic structure that resembles rings of the tail-tube protein in the assembled tail (Fig. 8b, c). The doubly-modified protein (delta-N and delta-HP) is primarily monomeric, closely resembling delta-N (Fig. 8a). Taken together, the structural and biochemical data suggest that the N-terminus facilitates assembly of the tail-tube protein, and that, in the precursor species, the β-hairpin negatively regulates the formation of rings. The location and amphipathic nature of

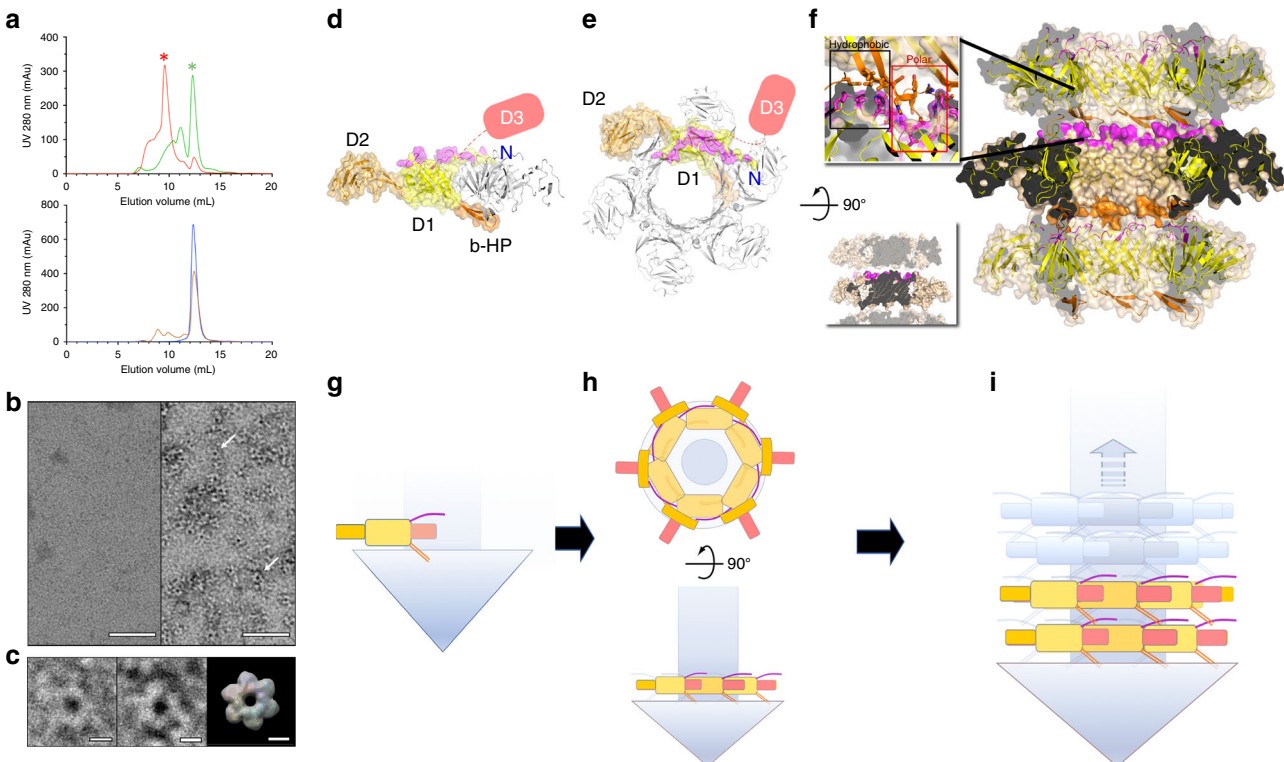

**Fig. 8 Assembly of the YSD1 phage tail. a** SEC chromatograms of YSD1_22 (green), a β-hairpin deletion mutant (delta-βHP, red), an N-terminal deletion mutant (delta-N, blue) and the double mutant (delta-βHP/delta-N, orange). Representative of three experiments from one (delta-βHP, double mutant) or two (YSD1_22, delta-N) purifications. The UV absorbance at 280 nm is displayed in milli-absorbance units (mAU). The asterisks indicate the peaks selected for negative-stain EM. The mass of delta-βHP was estimated to 231.7 kDa ± 4.2% by multi-angle laser light scattering. Theoretical mass: monomer = 38.7 kDa, hexamer = 232.2 kDa. **b** Representative negative-stain EM images of YSD1_22 and delta-βHP at ×67,000 magnification (two imaging sessions). **c** Zoomed view of ring-like objects highlighted by white arrows in **b**. For comparison, an hexameric ring of YSD1_22 is represented from the cryo-EM reconstruction of the assembled tail filtered to a resolution of 20 Å. Scale bars are 50 nm for **b** and 5 nm for **c**. **d**, **e** Orthogonal views of the hexameric rings forming the tail tube. For one subunit, domains 1, 2 and 3 and the β-hairpin (β-HP) are shown in yellow, brown, cyan and orange, respectively. Residues involved in inter-ring contacts are shown in magenta. Other subunits are coloured in grey and only one of these is shown in the side view of the ring. **f** A cross-section of the tail tube is shown as a ribbon within a brown molecular surface. Bottom-left inset: same view omitting β-HP. Top-left inset: zoom of the inter-ring interactions. Black and red boxes indicate contact areas with predominantly hydrophobic and polar interactions, respectively. **g–i** Model of the tail assembly process: Upon nucleation by the initiator (tail tip, represented as a blue triangle, or tail chaperones complex including the tape measure protein, represented as a blue cylinder), the N-terminus and β-HP form a splayed conformation (**g**), which is compatible with a stable hexameric ring (**h**). This conformation also positions the N-terminal arm and β-HP on either side of the rings making them available for axial polymerisation of the helical tail (**i**).

the β-hairpin are compatible with a role in directly or indirectly shielding the hydrophobic surfaces involved in inter-ring contacts (Fig. 6a and Supplementary Fig. 11b).

In the context of nucleation of the tail-tube polymerisation, the action of the initiator is thus expected to counteract the inhibitory role of the β-hairpin in the monomeric tail-tube protein. The details of these interactions, and indeed the identity of the initiator, are not known. Ultimately a splayed conformation between the N-terminus and the β-hairpin allows clipping in of each tail-tube protein subunits to build the hexameric ring (Fig. 8d, e, g, h). In this conformation the N-terminal arm faces the "headwards side" of the tail, while the β-hairpin points toward the tip of the tail. Together they brace the main β-sandwich of a neighbouring subunit (n-1). The N-terminal arm even reaches out to domain 2 of the following subunit (n-2) within a ring (Fig. 8e).

A splayed conformation of the β-hairpin and N-terminal arm not only allows ring formation but also make them accessible for inter-ring interactions, which are essential for axial extension of the tubular assembly to complete the morphogenesis of the tail tube (Fig. 8f, i). Unexpectedly, for such an elongated assembly subject to substantial shearing forces, there are relatively few inter-ring interactions. These interactions along the helical axis

rely almost exclusively on the extended β-hairpin, which contributes 93% of the inter-ring contact area. In the absence of the β-hairpin, protein interface analysis predicts stable rings but no further assembly. Only one inter-ring contact remains (Fig. 8f, insets; Supplementary Table 7). This mode of interaction is consistent with the characteristic flexibility of non-contractile tails, where inter-rings contacts may be disrupted without affecting the structural integrity of the rings, which remain bound to the central tape measure protein like beads on a string.

## Discussion

The architecture of the YSD1 head and tail components sheds light on assembly strategies and stabilising features that are conserved beyond the λ/χ-like phages. In the case of the head, an intricate non-covalent chainmail formed from auxiliary proteins was revealed, completing knowledge on phage capsids using the HK97 fold[18] (Supplementary Fig. 4). Equivalent stabilisation strategies of the λ/χ-like capsids are shared with herpesviruses highlighting a conserved requirement for reinforcement at the threefold vertices but also stapling interactions across the inter-capsomer seams.

In case of the tail component, the non-contractile tail of YSD1 is typical of the long tails found in *Siphoviridae*. The prevalence of this type of tails in the phage world remains intriguing given their intrinsic flexibility and inability to actively facilitate a contraction-driven insertion of the phage tail into the bacterial host. The features of the non-contractile tails raise three major questions that inspired this study: what induces the highly soluble precursors to polymerise into the tube, how is stability achieved in a flexible helical structure, and how does the structure facilitate DNA translocation to the interior of a bacterial cell? The structure of the YSD1 tail provides a molecular framework that will help address these points and reveals structural features likely to play a role in these processes.

The interior surface of the tube is lined with tracks of acidic residues along the length of the tail. These tracks present negatively charged motifs with a longitudinal spacing that matches the theoretical sizes of the minor and major grooves of viral DNA. Apparent conservation of these motifs in phage T5 suggests that they may have a role in guiding DNA during egress to facilitate its passage through the long and narrow conduit of the tail.

Compared to the other phage families, the *Siphoviridae* tails are characterised by their length and flexibility. The basic structural units in these tails are stable rings that have a compact hexameric organisation as anticipated from homologous tubular structures. By contrast, the interface between the rings involves surprisingly few contacts, relying almost exclusively on one structural element, the β-hairpin. This architecture allows for a pliable structure tolerant of local disruption. Morphogenesis of the tail is a highly regulated process in which the tail-tube protein β-hairpin appears to have antagonist roles. On one hand, it is a central feature of the tail that mediates all but one of the inter-ring contacts in the assembled tail structure; on the other hand, it restricts the self-assembly of the precursor protein into hexameric rings, raising the question whether it may be targeted in the nucleation of the tail assembly.

Given that sequence features in the capsid proteins and in the tail-tube proteins can now be identified in phage genome sequences as those that would impart increased stability, this study provides information to assist in selecting phages that might be superior for therapeutic and industrial applications, with prospects for longer shelf-life as infective reagents.

## Methods

**Isolation and amplification of YSD1.** From environmental water samples taken during a phage survey of the waterways of Cambridge UK, the phage YSD1 was isolated using the attenuated *Salmonella enterica* serovar Typhi BRD948[44]. The genome of YSD1 is available under accession number LR026998. The nucleotide sequence of YSD1 is 97% identical to χ phage, and there is 100% sequence identity at the protein level for major capsid protein, the auxiliary protein and the tail-tube protein[24]. Phage infections were performed in liquid media. Typically, 10 µL of serially diluted YSD1 was added to 200 µL of bacterial culture and incubated at 37 °C for 20 min to allow phage adsorption. While *Salmonella* Typhi BRD948 was used for YSD1 isolation, for reasons of biosafety, subsequent scale-up of phage production used *Salmonella* Typhimurium SL3261 *ΔfljB* (Supplementary Fig. 1b) in 14 cm culture dishes. Briefly, 60 µL of YSD1 phage preparation ($10^{-4}$ dilution) was added to 500 µL of an overnight culture of *S. Typhimurium* SL3261 *ΔfljB* and incubated for 20 min at 37 °C in two batches of 25 infections. Ten millilitres of soft agar (LB containing 0.35% noble agar) was added to the culture and poured using the double agar layer method and incubated overnight at 37 °C.

**Purification of YSD1 particles.** Ten millilitres of SM buffer (100 mM NaCl, 8 mM MgSO₄, 10 mM Tris pH 7.5) was added to each plate and incubated at room temperature for 10 min. The soft agar layer was scraped off using a disposable spreader and chloroform was subsequently added (1% v/v) to lyse the bacterial cells to release YSD1. The sample was then shaken vigorously, and agar and bacterial cell debris were pelleted at $11,000 \times g$ for 40 min (10 °C). The supernatant containing YSD1 were collected and DNase (1 µg mL⁻¹) and RNase (1 µg mL⁻¹) was then added to the supernatant and incubated for 30 min on ice. NaCl (1 M final concentration) was added and incubated on ice for 1 h with gentle mixing. Phage were precipitated from the media by adding polyethylene glycol (PEG) 8000

(10% w/v) and incubated at 4 °C overnight. Precipitated phage particles were collected by centrifugation at $11,000 \times g$ for 20 min at 4 °C and resuspended in SM buffer to 1.6% of the starting volume. An equal volume of chloroform was added to the resuspended bacteriophage suspension to remove residual PEG and cell debris and vortexed for 30 s. The organic and aqueous phases were separated by centrifugation at $3000 \times g$ for 15 min at 4 °C.

The aqueous phase containing YSD1 was removed and added to CsCl (0.05% w/v of bacteriophage suspension) and mixed gently to dissolve the CsCl. The suspension was layered onto a discontinuous CsCl gradient (2 mL of 0.17% w/v, 1.5 mL of 0.15% w/v and 1.5 mL of 0.145% w/v in SM buffer) in a Beckman SW41 centrifuge tube. Gradients were centrifuged at $83,000 \times g$ for 2 h (4 °C). Phage particles were collected from the gradient by piercing the side of the centrifuge tube with a syringe and removing the visible band in the gradient. Residual nucleic acid was removed from the phage preparation using floatation gradient centrifugation. Equal volumes of phage suspension (500 µL) and 7.2 M CsCl SM buffer were mixed and added to the bottom of a Beckman SW41 centrifuge tube. CsCl solutions (3 mL of 5 M and 7.5 mL of 3 M) were overlaid on top of the YSD1 sample and centrifuged at $83,000 \times g$ for 2 h (4 °C). Phage particles were collected (~500 µL) using a syringe as described above. CsCl was dialysed out of the phage stock twice with 2 L of SM buffer overnight at 4 °C. For EM analysis, 3 µL of sample was incubated on a freshly glow-discharged carbon-coated Formvar grid for 30 s. The sample was blotted from the grid and stained with 3 µL of 5% ammonium molybdate for a further 30 s and blotted again. Grids were imaged using a 120 kV FEI Spirit BioTwin with a Tietz F4.15 camera.

**Expression and purification of YSD1_17 and YSD1_22.** To produce sufficient major capsid protein (YSD1_17) and tail-tube (YSD1_22) proteins for structural analysis, the open-reading frames encoding YSD1_17 or YSD1_22 were synthesised (Genscript) with a C-terminal hexa-histidine tag, cloned into the protein expression vector pET21a (Novagen) and transformed into *Escherichia coli* C41 (DE3). YSD1_22 mutant constructs were prepared with deletions of key regions: delta-Nter, in which residues 1–8 were removed; delta-HP, in which residues 45–69 were removed and replaced with a glycine residue; and a double deletion mutant, containing both of the aforementioned modifications. The genes were synthesised (GeneWiz) with a C-terminal hexa-histidine tag and cloned into the protein expression vector pET21a (Novagen). The YSD1_22 mutant constructs were transformed into *E. coli* BL21 (DE3) cells. Transformed cultures were grown in Terrific broth (1.2 % w/v tryptone, 2.4 % w/v yeast extract, 0.4% v/v glycerol, 17 mM KH₂PO₄ and 72 mM g K₂HPO₄) for native proteins and Lysogeny Broth for the YSD1_22 mutants at 37 °C until cultures reached an optical density (OD600) of 0.8 and protein expression was induced with 0.5 mM isopropyl ß-ᴅ-1-thiogalactopyranoside. The cultures were then incubated with shaking overnight at 18 °C. Bacterial cell pellets were collected and lysed in lysis buffer (50 mM Tris pH 8.0, 400 mM NaCl, 2 mM MgCl₂ and 20 mM imidazole) using an Avestin cell press (three passes) for native proteins or by sonication for the YSD1_22 mutants. Purification of His-tagged proteins proceeded through Ni-affinity chromatography, with lysis buffer used for binding the proteins to a 5 mL nickel HisTrap HP column (GE Healthcare). After extensive washing with lysis buffer, each protein was eluted from the column with Elution buffer (50 mM Tris pH 8.0, 400 mM NaCl and 1 M imidazole). YSD1_17 and YSD1_22 were further purified by SEC using a HiLoad 16/600 Superdex 200 pg column (GE Healthcare) equilibrated in SEC buffer (25 mM Tris pH 8.0 and 200 mM NaCl). The YSD1_22 mutants were analysed by SEC using a ENrich™ SEC 650 10/300 (Bio-Rad) column equilibrated in SEC buffer. To estimate the protein molecular masses, we utilised In-line measurements of the refractive index and multi-angle laser light scattering (SEC-MALS) using DAWN HELEOS-II and Optilab T-rEX detector modules. ASTRA6 (Wyatt) was used for calculation of the molar mass.

**Negative staining electron microscopy of YSD1_22 mutants.** Five microlitres of purified protein was incubated on the top of freshly glow-discharged carbon-coated 200-mesh copper grids (PELCO) for 30 s at room temperature. Excess solution was removed using the Whatman filter paper. The grids were briefly rinsed three times in drops of ultrapure water and blotted after each rinse. The grids were incubated in a uranyl acetate solution (1% w/v) for 30 s. Excess stain was removed by blotting and the grids were dried at room temperature. The grids were imaged on a 120 keV Tecnai Spirit G2 microscope (FEI) equipped with a 4 K FEI Eagle camera.

**Small angle X-ray scattering.** SAXS was performed using Coflow SEC-SAXS at the Australian Synchrotron. Purified YSD1_22 was analysed at a pre-injection concentration of 10 mg mL⁻¹. Scattering was utilised over a $q$ range of 0.0–0.31 Å⁻¹ (Supplementary Fig. 10). A buffer blank for each SEC-SAXS run was prepared by averaging 10–20 frames pre or post protein elution. Scattering curves from peaks corresponding to YSD1_22 were then buffer subtracted and scaled across the elution peak, and compared for inter-particle effects. Identical curves (5–10) from elution were then averaged to provide curves for analysis. Data were analysed using the Primus, Scatter and Dammif modeller within the ATSAS Package (v. 3.0.2)[45]. To create the full model of YSD1_22 for comparison with solution scattering data, domain 3 was modelled using the Phyre2 webserver (http://www.sbg.bio.ic.ac.uk/~phyre2)[46] and appended to the C-terminus of the YSD1_22 structure derived

from cryo-EM and energy minimisation was performed in Chimera[47]. This model was fitted to the SAXS data by NMA performed using Crysol/Srefelx tool from the ATSAS package[48].

**X-ray crystallography**. Purified YSD1_17 at a concentration of 10 mg mL$^{-1}$ was screened for crystallisation in ~600 conditions, using commercially available screens. The protein crystallised rapidly and readily in many conditions. A condition from the Morpheus screen containing PEG3350 (4% w/v), glycerol (20% v/v), NaBr (0.2 M) and MOPS-HEPES buffer (0.1 M pH 7.5) was selected for optimisation. Crystals from this condition were directly harvested from sitting drops and flash frozen in liquid N$_2$ to 100 K. Data were collected at the Australia Synchrotron, indexed and scaled using the XDS package (v. 20151015), and merged with Aimless (v. 0.7.3) from the CCP4 package[49,50]. Initial phases were obtained by molecular replacement using PHASER (v. 2.6.0)[51] with the model of distantly related (27% amino acid identity) putative prophase protein 3BQW. Two independent molecules are present in the asymmetric unit and were refined with non-crystallography restraints using BUSTER (v. 2.10.3)[52] and built using COOT (v. 0.8.9.1)[53]. Residues 4–93, 108–231, 238–258, 261–277, 279–357 were modelled. The two copies superpose with an all-atom RMSD of 0.75 Å over 2586 atoms.

**Cryo-electron microscopy**. An aliquot of 5 µL of purified YSD1 particles was applied to a glow-discharged R2/2 holey carbon grid (Quantifoil Micro Tools GmbH, Germany). The grid was blotted for 2 s (−3 blot force, 0 s drain, 100% humidity) and plunge-frozen in liquid ethane using a Vitrobot Mark IV (FEI/Thermo Fisher Scientific). Grids were transferred under liquid nitrogen to a Titan Krios transmission EM (FEI/Thermo Fisher Scientific) operated at 300 kV and set for parallel illumination. Movies were recorded using SerialEM (v. 3.6) at a defocus range of 0.5–2.5 µm on a K2 Summit direct electron detector (Gatan Inc., USA) in super-resolution mode at a nominal magnification of ×105,000 (a pixel size of 0.67 Å at the specimen level) and using energy filtering. Each movie consisted of a 12 s exposure divided into 30 frames, with a total dose of 27.24 e$^-$ Å$^{-1}$.

**Image processing**. The movies were binned two times by Fourier cropping before motion correction and integrated with MotionCor2 (v1.1.0)[54], giving a final pixel size of 1.34 Å. Simultaneously, the antistrophic magnification distortion was corrected in MotionCor2 using values obtained with the mag_distortion_estimate programme (1.4%, 61.1 degrees). The contrast transfer function (CTF) parameters of each image were determined using CTFFIND (v. 4.1.8)[55] and images with significant astigmatism or drift were removed. A total of 1489 micrographs were used for particle picking and 3D reconstruction.

**Single-particle reconstruction of YSD1 capsid**. Using RELION (v. 2.1), capsids were selected from micrographs and used to generate a template for autopicking. The coordinates were boxed, and 2D classification was performed to select 7366 intact capsids. Using a soft sphere as an initial model, 3D classification with icosahedral symmetry applied was used to identify the best particles for 3D reconstruction. The orientations and alignments of 5449 particles were refined to produce a 3.9 Å reconstruction (FSC = 0.143), after post-processing with a softened mask. For the major capsid protein, the crystal structure of YSD1_17 was used as an initial model. For the auxiliary protein, we used the structure of the SHP protein from lambdoid phage 21 (PDB ID 1TD0) as an initial model which had a 32.6% sequence identity over 43 residues. Modelling was performed in Coot and the models were refined in real space with the phenix.real_space_refine programme (PHENIX v. 1.13)[56].

**Helical reconstruction of YSD1 tail**. Tails were boxed from the micrographs as overlapping segments using e2helixboxer.py from the EMAN2 suite (v. 2.12)[57] and extracted with an interbox distance of approximately one helical rise (~40 Å). Two dimensional (2D) classification of the 184,501 particles into 50 classes using RELION[58] was used to clean the dataset, resulting in 147,809 particles. The initial 3D reconstruction was performed using a featureless cylinder and the helical parameters (twist of 19.7 degrees, rise of 42.7 Å) determined using SPRING (v. 0.86.1661)[59] (Supplementary Fig. 6), from which an initial model was obtained that was subsequently used during 3D refinement. C6 symmetry was imposed during the reconstruction. 3D refinement of the particles using a local search of the helical symmetry refined the helical parameters to a twist of 19.7° and a rise of 41.2 Å. Every second frame of the movies was averaged with the previous frame and the particles were re-extracted for movie processing and particle polishing. The final 3D refinement yielded a 3.5 Å-resolution reconstruction (FSC = 0.143), after post-processing with a softened mask. The final map was filtered according to the local resolution using RELION. ROSETTA (v. 1.16) was used to generate an initial model. The sequence was assigned to the chain and the model was rebuilt using Coot and refined using phenix.real_space_refine.

**Model analysis**. PyMOL (The PyMOL Molecular Graphics System, version 2.0.6; Schrödinger, LLC) and UCSF Chimera (v. 1.13.1)[47] were used to render images of the structures. The surface potential and electrostatics were calculated with the adaptive Poisson–Boltzmann solver[60] as implemented in PyMOL with hydrogen atoms added with the pdb2pqr method. The binding surfaces were calculated by using the Proteins, Interfaces, Structures, and Assemblies server[61]. The Consurf server was used with default parameters to estimate evolutionary conservation scores and map them onto the tail tube structure[62]. Homologue search using HMMER identified 91 unique sequences with 35–95% sequence identity. Residues in Consurf categories 7–9 (most conserved) are shown in Fig. 5.

**Reporting summary**. Further information on research design is available in the Nature Research Reporting Summary linked to this Article.

## Data availability
Data supporting the findings of this manuscript are available from the corresponding authors upon reasonable request. A Reporting Summary for this Article is available as a Supplementary Information file. Source data are provided with this paper. The coordinates and diffraction data of the major capsid protein crystal structure have been deposited in the Protein Data Bank (YSD1_17; PDB 6XGP). The coordinates for the capsid (YSD1_16 and YSD1_17; PDB 6XGQ) and tail tube (YSD1_22; PDB 6XGR) from the cryo-EM structure of the mature YSD1 phage have been deposited in the Protein Data Bank. The icosahedral reconstruction of the capsid (EMD-22183) and helical reconstruction of the tail (EMD-22182) have been deposited to the Electron Microscopy Bank. SAXS scatter and envelope data for YSD1_22 have been deposited in the Small angle scattering data bank (SASDB ID: SASDJM2).

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

## Acknowledgements

We thank Iain Hay for expert assistance with artwork. Cryo-EM imaging was performed at the Ramaciotti Center for Cryo-Electron Microscopy at Monash University. We acknowledge the Australian Synchrotron for access to beamlines MX1 and MX2 for X-ray crystallographic analysis (CAP11027) and SAXS-WAXS for X-ray solution scattering (M12480), the Monash Molecular Crystallisation Facility for their assistance crystallographic screening and optimisation, and the MASSIVE HPC facility for computing resources. Research was supported by Programme Grant 1092262 from the National Health and Medical Research Council of Australia (NHMRC). R.G. was a Sir Henry Wellcome Fellow (106077/Z/14/Z), T.L. was an Australian Research Council Laureate Fellow (FL130100038), F.C. was an Australian Research Council Future Fellow (FT0100893).

## Author contributions

J.M.H., R.A.D., R.G., J.W., M.J.B. and D.P. designed and carried out analysis. H.V., M.J.B., G.D., F.C. provided expertise to analyses. F.C. and T.L. supervised experimental work and evaluated data. J.M.H., R.A.D., T.L. and F.C. evaluated results and wrote the manuscript.

## Competing interests

The authors declare no competing interests.
