## [Peer Review File · Nature Communications]

Peer Review File, Reviewers' comments first round:

Reviewer #1 (Remarks to the Author):

Comments on "The architecture and stabilisation of flagellotropic tailed bacteriophages"

The authors present combined structural methods including cryoEM, crystallography, and bioSAXS to provide novel insights into the structure and potential assembly of the previously poorly characterized but fascinating flagellotropic phage family, specifically the non-contractile tails therein. The resultant data delineate clearly the structural features underlying assembly and suggest reasons that these structures are able to withstand the substantial shear forces experienced when bound to the rotary extracellular flagellar appendages of their bacterial host. The model of DNA passage is compelling and a foundation for further studies probing understanding of rapid passage of these large genomic substrates from phage to the bacteria they infect. Phage reagents are a hot bed of potential therapeutic applications currently and collectively I believe this work will be of significant and broad interest to those studying protein assemblies and infection.

Line 76 – "Recent genome sequence analysis had indicated that, at a protein structural level, YSD1 and χ -phage virions are indistinguishable²²" –had to check the ref to see that they mean the protein seq would be 99-100% identical (however structures can be highly similar even with disparate sequence). This should be rephrased for clarity.

Line 78 and throughout the manuscript please define resolution values as such "3.8 Ang structure" should be "3.8 Ang resolution structure"

Figure S1a – please provide a scale in the negative stain image as well as a brief mention in the methods (stain type etc).

Line 129, add resolution of the crystal structure here. Why is the crystal structure captured as a monomer? Please specify number in the au in the methods as well as any potential crystal packing interfaces of relevance. Are any mimicking that of the cryo EM assembly? Or alternatively any potential influence of the observed structure from crystal packing?

Line 154, cyan should be magenta?

Line 167, here and throughout the manuscript – please provide the number of atoms used in the rmsd calculation

Line 220; the values to the hundredths place given the potential coordinate error at this resolution are likely unwarranted.

Line 329 – "The β -hairpin, the N-terminal arm and loops on the head-proximal side of the hexameric ring protrude from the SAXS envelope (Fig. 7a) suggesting that these structural features are either flexible or folded back onto the core structure" – I think that these are certainly options, but even if these small extensions were rigid, I'm not sure that density would be visible within the resolution scope of bioSAXS. In general, although the model of assembly is compelling, I think the statements in this section (lines 328-355) are reaching beyond the scope of the data. Some mutagenesis or additional experimentation could be nice in this regard to solidify this proposed assembly. The comment above also applies again in the Discussion (lines 395-397), and Figure 7.

Why was RELION2 used for the processing? It might be appropriate to reprocess in RELION3.1, particularly to see if any additional insight can be pulled from the region of the helical tail that is connecting to D3.

A SAXS stats table is not present in the supplement – some modified form of what is in the guidelines here: <https://journals.iucr.org/d/issues/2017/09/00/jc5010/> would be appropriate.

Throughout: capsomer should be capsomere?

Reviewer #2 (Remarks to the Author):

The MS "The architecture and stabilisation of flagellotropic tailed bacteriophages" describes high resolution structure of bacteriophage YSD1 that infects Salmonella Typhi. YSD1 is similar to phage Chi, which is known to attach to the bacterium flagellum. The MS contains a substantial number of bold statements, but most of them, unfortunately, remain unsupported by the findings. For this reason, the MS needs to be substantially modified to become an actual article. One positive aspect of the MS is that the figures are absolutely beautiful.

Please find my line-by-line comments below.

The abstract

- However, it is unclear how tailed bacteriophages withstand the substantial rotary shear forces generated as they move down the spinning flagellum.

I am not sure why the "rotary shear forces" are singled out here. Are the "shear forces" acting on the particle attached to cell surface lipopolysaccharides with its tail fibers weaker?

- In addition, the structure of the tail reveals concerted rearrangements

A supporting evidence for concerted rearrangement is not presented in this paper.

- provide an elegant means for promoting genome translocation

To claim that something promotes something, you need to show evidence that in the absence of this feature the process is inhibited.

The Intro

- In contractile tails of myoviruses, an outer sheath adds to the stability of the central tail-tube, however it is unclear how tail-tube integrity is maintained in flexible tails found in the majority of phages.

What is the basis for this remarkable statement? Myophage tail tubes are pretty stable in solution (although the baseplate is required for assembly). In the contracted state, about half of the tube protrudes below the plane of the baseplate. The sheath does NOT interact with the tube in this state at all, so the sheath is not required to maintain the stability of the tube for DNA translocation. You really need to read some literature...

- The best available model of such a tail is derived from a pseudo-atomic model for phage T5, which has an atypical 3-fold symmetry.

The structure is nearly perfectly 6-fold symmetric. There is no need to bash it for being 3-fold.

- In the absence of a contractile sheath, how the ~10 µm (~60 kbp) genome efficiently transits through this long narrow tunnel is also not fully understood.

Again, this is a remarkable statement in its erroneousness. DNA translocation and sheath contraction are not linked.

- and four classes of chainmail

I think there is only one type of chainmail, which is actually cross-linked. All other cases of capsid stabilizations are not chainmail. I believe there are only three fundamentally different ways to stabilize the capsid: 1) covalent cross-linking of capsomers (chainmail), 2) additional domains to the HK97-fold and 3) additional decoration proteins. 1), 2) and 3) can exist in different combinations (e.g. 2) + 3) or 1) +3), etc.). Prolate or not has nothing to do with "chainmail stabilization" (which is an incorrect concept as explained above) as far as existing data show.

- (ii) the P22-like capsids that are stabilised by an inserted domain in the minimal HK97 fold19; The term "insertion domain" was introduced a few years before Kristin Parent's publication

(PMID: 15878991). It is a pity that Kristin failed to acknowledge that the insertion domain in the P22 capsid protein is located at the same position as the insertion domain in T4 gp23/gp24 (compare Fig. 2 from Fokine et al PNAS 2005 and Fig. 5A from Parent et al Structure 2010). And I find it even more disappointing that the discovery of the insertion domain and its possible role in capsid stabilization has not been attributed to the original authors in this manuscript.

The Results

The internal surface of the tail-tube facilitates DNA ejection

- Given the conservation of similar acidic residues dyads in the T5 crystal structure and sequences of tail-tube proteins, we suggest that this DNA translocation mechanism is general to all Siphoviridae.

Are those residues conserved? Can you please support this statement by showing sequence conservation mapped onto the structure?

Assembly of a flexible yet robust tail-tube

- These changes would mask the specific regions that form the inter-ring interfaces in the assembled tail. In a model of nucleation of the tail-tube polymerisation summarised in Figure 7, the action of the initiator would thus require splaying of the β -hairpin and N-terminal arm to allow clipping of tail-tube protein subunits into the hexameric ring.

So, we cannot see a density for the hairpin or the N-terminal arm in the SAXS envelope and the conclusion is that the two are folded back and interact with each other? This conclusion needs to be supported by additional experiments such as cross-linking or mutagenesis that would show that the two elements interact in the recombinant protein. As this finding is central to the later stated hypothesis, the hypothesis has a very shaky foundation.

Discussion

- These features raise three major questions addressed in this study: what induces the highly soluble precursors to polymerise into the tube, how is stability achieved in a flexible helical structure, and how does the structure promote DNA translocation?

This study reported the structure and atomic models of the capsid and tail tube of phage YSD1.

The first question is about particle morphogenesis and it cannot be answered with the structure data alone. It requires hypotheses, mutagenesis and analysis of assembly intermediates. The

second question is about stability. To claim stability, the actual stability needs to be measured.

Mutants have to be created and their stability needs to be measured. The third question is about DNA translocation. Finding a negatively charged channel tells us nothing about the mechanism.

The portal protein of phage phi29 was the first phage-derived negatively charged channel. Please show how the YSD1 is different. What do those regularly spaced residues in the YSD1 channel do?

- Here we show that to form the hexameric building blocks of the tail, the β -hairpin and N-terminal tail need to unmask complementary hydrophobic regions in the tail tube protein.

Please see my earlier remarks about the actual evidence of the claimed property.

- More than this, the precise arrangement of negative charges revealed here suggests a mechanistic explanation for the theoretical need for DNA to ratchet from the tube once the initial impetus to the DNA from the pressurized environment of the capsid begins to wane.

I am not sure that I follow the logics. The tube is negatively charged. It contains regularly spaced negatively charged amino acids as it should (it is a repetitive polymeric structure). All that repels the DNA and does not let it stick to the side of the tube. What ratchets are we discussing here?

The architecture and stabilisation of flagellotropic tailed bacteriophages

Hardy, Dunstan et al.

We thank the reviewers for their insightful comments and suggestions that helped improve the manuscript. We provide a point-by-point response to the reviewers' points as follows.

Reviewer #1:

Comments on "The architecture and stabilisation of flagellotropic tailed bacteriophages"

The authors present combined structural methods including cryoEM, crystallography, and bioSAXS to provide novel insights into the structure and potential assembly of the previously poorly characterized but fascinating flagellotropic phage family, specifically the non-contractile tails therein. The resultant data delineate clearly the structural features underlying assembly and suggest reasons that these structures are able to withstand the substantial shear forces experienced when bound to the rotary extracellular flagellar appendages of their bacterial host. The model of DNA passage is compelling and a foundation for further studies probing understanding of rapid passage of these large genomic substrates from phage to the bacteria they infect. Phage reagents are a hot bed of potential therapeutic applications currently and collectively I believe this work will be of significant and broad interest to those studying protein assemblies and infection.

1. Line 76 – "Recent genome sequence analysis had indicated that, at a protein structural level, YSD1 and χ -phage virions are indistinguishable²²" –had to check the ref to see that they mean the protein seq would be 99-100% identical (however structures can be highly similar even with disparate sequence). This should be rephrased for clarity.

We have rephrased the statement as follows: "Comparison to the genome sequence of the χ -phage indicated that YSD1 and χ -phage share structural proteins with identical amino acid sequences²²."

2. Line 78 and throughout the manuscript please define resolution values as such "3.8 Ang structure" should be "3.8 Ang resolution structure"

We have made these corrections. Five occurrences were modified.

3. Figure S1a – please provide a scale in the negative stain image as well as a brief mention in the methods (stain type etc).

We have added the scale bars to Fig. S1a and the METHODS section has been revised to include details of TEM p.19.

4. Line 129, add resolution of the crystal structure here.

Done.

5. Why is the crystal structure captured as a monomer? Please specify number in the au in the methods as well as any potential crystal packing interfaces of relevance. Are any mimicking that of the cryo EM assembly? Or alternatively any potential influence of the observed structure from crystal packing?

In the absence of other viral proteins (e.g. the auxiliary protein), the recombinant MCP is produced as a monomeric species as assessed by size-exclusion chromatography. Two copies of the monomeric form are found in the asymmetric unit of the crystal structure. The two copies have the same compact conformation (all atom rmsd of 0.70 Å). Since the molecules have

different crystal contacts, the conformation observed in both independent MCPs of the asymmetric unit is unlikely to be caused by crystal packing. In one of the two molecules, only one residue of the N-terminus and E-loop respectively is within 5 Å of a symmetry-related molecule. In this molecule, the N-terminus and E-loop point toward cavities in the crystal that could accommodate extended conformations.

We agree with the reviewer that these are important points. We have modified Table S2 and added the following text:

p.5 – “These rearrangements are unlikely to be caused by crystal packing since the same compact conformations are seen for both of the independent molecules in the asymmetric unit of the crystal despite completely different environments.”

and:

p. 20 - “Two independent molecules are present in the asymmetric unit and were refined with non-crystallography restraints using BUSTER v. 2.10.2⁵¹ and built using COOT⁵². Residues 4-93, 108-233, 240-259, 264-357 were modelled. The two copies superpose with an all-atom RMSD of 0.70 Å over 5212 atoms.”

Unfortunately, contacts in the crystal do not mimic the cryo-EM assembly in any meaningful way.

6. Line 154, cyan should be magenta?

Done.

7. Line 167, here and throughout the manuscript – please provide the number of atoms used in the rmsd calculation

Done. Details have been added in p. 6 for the Auxiliary protein (“RMSD between Ca of 88 equivalent residues of 1.7 Å), p. 20 l. 566-567 and Table S3-S7.

8. Line 220; the values to the hundredths place given the potential coordinate error at this resolution are likely unwarranted.

We have rounded the decimals accordingly.

9. Line 329 – “The β-hairpin, the N-terminal arm and loops on the head-proximal side of the hexameric ring protrude from the SAXS envelope (Fig. 7a) suggesting that these structural features are either flexible or folded back onto the core structure”

I think that these are certainly options, but even if these small extensions were rigid, I’m not sure that density would be visible within the resolution scope of bioSAXS. In general, although the model of assembly is compelling, I think the statements in this section (lines 328-355) are reaching beyond the scope of the data. Some mutagenesis or additional experimentation could be nice in this regard to solidify this proposed assembly. The comment above also applies again in the Discussion (lines 395-397), and Figure 7.

We agree and have modified the statements in the results (now p.13 l. 341-345), the discussion and removed panel 7e, which interpreted the SAXS data as a schematic model. The SAXS data (old panel 7a) is now presented in supplementary material S11a together with additional modelling and comparison with the NMR structure of the tail tube protein of phage lambda (S11b-c).

We appreciate the point regarding a mutagenesis-based project to test details of the non-covalent linkages in the hexameric rings, and have considered this previously. Unfortunately, with a lytic phage such as YSD1, we have no way to test for the assembly of mutant structural proteins into virions. However, we confirmed contrasting roles of the beta-hairpin and N-terminus in self-assembly propensity by deletion analysis in the recombinant tail tube protein. This data is presented in Fig. 7a, b and discussed p. 13 l. 347-361.

10. Why was RELION2 used for the processing? It might be appropriate to reprocess in RELION3.1, particularly to see if any additional insight can be pulled from the region of the helical tail that is connecting to D3.

We have reprocessed the data with RELION 3.1. However, in this case, the improvements to our reconstructions were negligible.

Per-particle CTF correction improved the resolution of the capsid by 0.1 Å and in the case of the tail reconstruction, it introduced additional error which actually worsened the resolution. The estimation of the magnification anisotropy from the data also did not improve the reconstruction more than our manual measurements of the distortion. There was no improvement to density of the C-terminal domain in the reconstruction of the tail.

11. A SAXS stats table is not present in the supplement – some modified form of what is in the guidelines here: <https://journals.iucr.org/d/issues/2017/09/00/jc5010/> would be appropriate.

The table was added as Table S11.

12. Throughout: capsomer should be capsomere?

Both terms are accepted by the Collins or Merriam-Webster dictionaries. We prefer capsomer.

Reviewer #2:

The MS “The architecture and stabilisation of flagellotropic tailed bacteriophages” describes high resolution structure of bacteriophage YSD1 that infects Salmonella Typhi. YSD1 is similar to phage Chi, which is known to attach to the bacterium flagellum. The MS contains a substantial number of bold statements, but most of them, unfortunately, remain unsupported by the findings. For this reason, the MS needs to be substantially modified to become an actual article. One positive aspect of the MS is that the figures are absolutely beautiful.

Response: we have substantially modified the manuscript and thank the Reviewer for the suggested improvements. Wherever possible, we have addressed the Reviewer’s reservations by further analysis or experimentally (cf. new Fig. 3c,d and 5a,b; Fig. S9 and S11), and otherwise removed or clarified the relevant statements.

Please find my line-by-line comments below.

The abstract

1. *However, it is unclear how tailed bacteriophages withstand the substantial rotary shear forces generated as they move down the spinning flagellum.*

I am not sure why the “rotary shear forces” are singled out here. Are the “shear forces” acting on the particle attached to cell surface lipopolysaccharides with its tail fibers weaker? We appreciate the query and do not single out the rotary component in the revised text. We have expanded the statement (Page 2, line 25-26) to make clear that the values previously calculated for the drag forces and torques are extraordinary for nanoscale particles (Ref. 10-Katsamba et al). It now reads “Given the substantial drag forces and torques that have been calculated to impact on them as they move down the spinning flagellum, it was of great interest to investigate the structural details of a flagellotropic bacteriophage.”.

We are not aware that anyone has calculated the “shear force” acting on a particle attached to cell surface lipopolysaccharides with its tail fibers, but one might expect given that this shear would be similar for a particle anywhere on the bacterial cell surface. Thus, for a particle attached to the flagellum, it would experience this force plus the calculated shear due to rotation of the flagellum and movement of the phage towards the cell surface. These shear forces would be the distinct environmental concern that the phage might have evolved to protect against.

2. *In addition, the structure of the tail reveals concerted rearrangements*

A supporting evidence for concerted rearrangement is not presented in this paper. We agree with the reviewer’s point. We have removed the term “concerted rearrangements” and completely rewritten this section.

3. *“provide an elegant means for promoting genome translocation”*

To claim that something promotes something, you need to show evidence that in the absence of this feature the process is inhibited.

We agree. It is reasonable to infer that these features facilitate genome translocation but the term “promote” has been removed. We have changed the text accordingly to: “...provide regularly spaced motifs well suited to facilitate DNA translocation into the bacterial host cell.”.

The Intro

4. *In contractile tails of myoviruses, an outer sheath adds to the stability of the central tail-tube, however it is unclear how tail-tube integrity is maintained in flexible tails found in the majority of phages.*

What is the basis for this remarkable statement? Myophage tail tubes are pretty stable in solution (although the baseplate is required for assembly). In the contracted state, about half of the tube protrudes below the plane of the baseplate. The sheath does NOT interact with the tube in this state at all, so the sheath is not required to maintain the stability of the tube for DNA translocation. You really need to read some literature...

We meant to refer to the stability of the tube in the entire extra-cellular stage of the phage, not specifically during DNA translocation. In most of the extra-cellular phase of *Myoviridae*, the contractile sheath that braces the inner tail tube form inter-molecular contacts and an external shell. These features are very likely to contribute to the stability of the assembly.

We appreciate the need to provide additional detail in the text in the Introduction, and have done so (page 3, lines 53-57) as follows:

“In contractile tails of myoviruses, an outer sheath braces the entire central tail-tube in the pre-contracted state, presumably adding stability to the assembly. Given that they have no outer tube structure and are up to twice as long, it is of interest to understand how tail-tube integrity is maintained in flexible tails found in flexible tails typical of the *Siphoviridae*.”

5. *The best available model of such a tail is derived from a pseudo-atomic model for phage T5, which has an atypical 3-fold symmetry.*

The structure is nearly perfectly 6-fold symmetric. There is no need to bash it for being 3-fold. Bash? The mention of the 3-fold symmetry was not intended to be negative. The T5 structure is indeed interesting and is discussed later in the text p. 9, l. 229, p. 12, l. 304-307 and Supplementary Figure 7 and 9 (also cf. response to comment 9).

6. *In the absence of a contractile sheath, how the ~10 μm (~60 kbp) genome efficiently transits through this long narrow tunnel is also not fully understood.*

Again, this is a remarkable statement in its erroneousess. DNA translocation and sheath contraction are not linked.

We have revised the text to remove the reference to the contractile sheath.

7. *and four classes of chainmail*

I think there is only one type of chainmail, which is actually cross-linked. All other cases of capsid stabilizations are not chainmail. I believe there are only three fundamentally different ways to stabilize the capsid: 1) covalent cross-linking of capsomers (chainmail), 2) additional domains to the HK97-fold and 3) additional decoration proteins. 1), 2) and 3) can exist in different combinations (e.g. 2) + 3) or 1) + 3), etc.). Prolate or not has nothing to do with “chainmail stabilization” (which is an incorrect concept as explained above) as far as existing data show.

In the revised manuscript, we have kept the broader definition of chainmail introduced by Hong Zhou and coll., which allows for non-covalent chainmail and defines 4 broad categories. We believe that it provides a useful way to describe our structure and compare it with the original HK97 structure and those of the more recent structures that used this terminology. To account for the reviewer’s reservations, we distinguish chainmail-like organisation from the *bona fide* covalent chainmail.

We agree with the reviewer’s categorisation of stabilisation strategies: we have removed the separate statement on prolate heads and added details on the insertion in T4 and decoration proteins.

Updated text p.3, l. 67-71: The HK97-like capsids assemble into a *bona fide* chainmail formed by a covalent crosslinking of the major capsid protein^{17, 18}. Extending the strict definition of protein chainmail to include chains that are non-covalently interlocked, three other classes of chainmail-like organisations have been defined¹⁹: (i) the T4 and P22-like capsids that are stabilised by an inserted domain in the minimal HK97 fold^{20, 21, 22...}”

and l. 73-74:

“In the absence of a covalent chainmail, additional proteins called decoration, auxiliary or cementing proteins may add to the stability of the particle.”

8. - (ii) the P22-like capsids that are stabilised by an inserted domain in the minimal HK97 fold¹⁹;

The term “insertion domain” was introduced a few years before Kristin Parent’s publication (PMID: 15878991). It is a pity that Kristin failed to acknowledge that the insertion domain in the P22 capsid protein is located at the same position as the insertion domain in T4 gp23/gp24 (compare Fig. 2 from Fokine et al PNAS 2005 and Fig. 5A from Parent et al Structure 2010). And I find it even more disappointing that the discovery of the insertion domain and its possible role in capsid stabilization has not been attributed to the original authors in this manuscript. The reference to Parent et al was provided as an example where the capsid does not have auxiliary proteins or chemical crosslinks. We have now added T4 to this sentence and included reference PMID: 15878991 with the original description of an insertion domain (cf. updated text in point 7 above).

The Results

9. *The internal surface of the tail-tube facilitates DNA ejection - Given the conservation of similar acidic residues dyads in the T5 crystal structure and sequences of tail-tube proteins, we suggest that this DNA translocation mechanism is general to all Siphoviridae.*

Are those residues conserved? Can you please support this statement by showing sequence conservation mapped onto the structure?

We thank the reviewer for the suggestion and the conservation analysis has been integrated into the manuscript as follows.

Two new figures have been provided to support this point (Fig. 5c-d and Supplementary Fig. 9). Strict conservation is not necessary for a motif matching non sequence-specific features in the viral DNA. Nevertheless, analysis of sequence conservation mapped onto the structure indicates positive selection at these sites (Fig. 5c-d). Structural comparison with T5 also shows a similar pattern (Asp/Glu residues close in space and/or sequence, and spacing between these motifs; Supp. Fig. S9a). Presence of these similarities despite different organisations - trimeric rings with a quasi 6-fold symmetry rather than hexameric rings – is suggestive of a conserved role.

Comparison with tail proteins for which a structure of the assembled tail is not known or cannot be modelled reliably is risky. Consensus sequences for each of the clusters defined in Fig. S7 have candidate motifs (Supp. Fig. S9b) but structural analysis will be required to validate their positioning and function.

The revised text reads (p. 12, l. 308-317):

“We can provide no experimental test of these observations, but note that similar acidic residues dyads are present in the T5 structure despite its different organisation based on trimeric rings with a quasi 6-fold symmetry rather than hexameric rings (Supplementary Fig. 9a-f). Mapping of evolutionary conservation onto the tail-tube protein structure identified residues E47, D243, D249, D250 as being highly conserved (Fig. 5c,d). Identification of these motifs in distantly related phages for which sequences cannot be aligned with YSD1_22 will require structure-based analysis to confirm their locations in the assembled tails. However, candidate motifs are present in all the tail-tube protein sequences analysed here (Supplementary Fig. 9g; Table S9) suggesting that this structural feature is general to all *Siphoviridae*.”

10. **Assembly of a flexible yet robust tail-tube**

- These changes would mask the specific regions that form the inter-ring interfaces in the assembled tail. In a model of nucleation of the tail-tube polymerisation summarised in Figure

7, the action of the initiator would thus require splaying of the β -hairpin and N-terminal arm to allow clipping of tail-tube protein subunits into the hexameric ring.

So, we cannot see a density for the hairpin or the N-terminal arm in the SAXS envelope and the conclusion is that the two are folded back and interact with each other? This conclusion needs to be supported by additional experiments such as cross-linking or mutagenesis that would show that the two elements interact in the recombinant protein. As this finding is central to the later stated hypothesis, the hypothesis has a very shaky foundation.

We appreciate this point regarding experimental validation of the structural role of the hairpin and N-terminal arm and have addressed it as follows.

a. We present modelling suggesting that these two structural elements are more dynamic in the monomeric state than the rest of the structure based on normal mode analysis combined with the SAXS data and published NMR data on the lambda tail protein (Supp. Fig. S11). We also note that “These differences with the cryo-EM structure could be due to the lack of resolution of the SAXS envelope” (p. 13, l. 343). By contrast, these elements are in a stable, splayed conformation in the assembled tail involved in multiple inter-subunit interactions.

As presented in the revised manuscript and described below, our analysis does not rely on the precise conformation of the β -hairpin and N-terminal arm in the monomeric tail tube protein or whether they interact together or not.

We have thus removed speculations regarding a folded back conformation of the N-ter arm and/or hairpin and Fig. 7e from the old manuscript. We now only note in p. 14, l. 363-365 that “The location and amphipathic nature of the β -hairpin are compatible with a role in directly or indirectly shielding the hydrophobic surfaces involved inter-ring contacts (Fig. 5a and Supplementary Fig. 11b).”

b. Importantly, we have added experimental data showing that deletion of the β -hairpin and N-terminal arm impact the self-assembly ability of the soluble tail tube protein. The SEC, MALLS and TEM data show that removal of the N-terminal increases the monodispersity of the monomeric species, while removal of the β -hairpin promotes self-assembly of ring-like structures. These experiments are presented in an updated Fig. 7 and a completely rewritten section (e.g. most of p. 13 and 14).

We have also refocused our discussion of the initiator’s role accordingly as follows:

“In the context of nucleation of the tail-tube polymerisation, the action of the initiator is thus expected to counteract the inhibitory role of the β -hairpin in the monomeric tail-tube protein. The details of these interactions, and indeed the identity of the initiator, are not known.”

11. Discussion

- These features raise three major questions addressed in this study: what induces the highly soluble precursors to polymerise into the tube, how is stability achieved in a flexible helical structure, and how does the structure promote DNA translocation?

This study reported the structure and atomic models of the capsid and tail tube of phage YSD1. The first question is about particle morphogenesis and it cannot be answered with the structure data alone. It requires hypotheses, mutagenesis and analysis of assembly intermediates. The second question is about stability. To claim stability, the actual stability needs to be measured. Mutants have to be created and their stability needs to be measured.

The DISCUSSION has been rewritten to more clearly distinguish the questions that inspired the study from those that we have addressed with direct experiments (p. 16, lines 426-431). Additional experiment and further analysis have been integrated into the revised DISCUSSION.

The discussion focuses on the novel structural insights provided by this study. As far as the tail is concerned, the resolution of the structure allows, for the first time, a detailed description of features in the assembled tube. In the absence of reverse-genetics or tail assembly system for YSD1, we have further supported our interpretations of the role of the beta-hairpin and N-terminus by additional modelling, structural analysis (e.g. sequence conservation), and new experiments.

For the first point (morphogenesis), we agree with the reviewer that the structural comparison of SAXS envelope and cryo-EM structure alone is not sufficient to implicate the hairpin and/or N-terminus as regulatory elements in the tail tube protein self-assembly. We have added support for conformational variability of these regions (Supp. Fig. S11) and have toned down structural interpretations. Importantly, we add support to the functional role of these elements in self-assembly by deletion mutagenesis analysed by SEC, MALLS and TEM (Fig. 7). Further validations in systems where the phage can be readily modified, or a complete in vitro assembly system is available will undoubtedly be of value but are beyond the scope of this study.

For the second point (stability), the role of the beta-hairpin in inter-ring stability is inferred from buried surface analysis and estimation of free energy of the various complexes (Fig. 7e and PISA data in Table S7). Such analysis is well validated for protein complexes and we do not see a requirement for biophysical experiments in such a clear case (only 1 predicted inter-ring contacts in the absence of the hairpin). To define the specific determinants of assembly at the level of specific amino acid, fine mutagenesis and biophysical assessment of stability will be most interesting. We are not in a position to perform these but, in the revised manuscript, we suggest future avenues from a mapping of sequence conservation (new panels c,d in Fig. 5).

The revised text reads:

“By contrast, the interface between the rings involves surprisingly few contacts, relying almost exclusively on one structural element, the β -hairpin. This architecture allows for a pliable structure tolerant of local disruption. Morphogenesis of the tail is a highly regulated process in which the tail-tube protein β -hairpin appears to have antagonist roles. On one hand, it is a central feature of the tail that mediates all but one of the inter-ring contacts in the assembled tail structure; on the other hand, it restricts the self-assembly of the precursor protein into hexameric rings, raising the question whether it may be targeted in the nucleation of the tail assembly.”

The third question is about DNA translocation. Finding a negatively charged channel tells us nothing about the mechanism. The portal protein of phage phi29 was the first phage-derived negatively charged channel. Please show how the YSD1 is different. What do those regularly spaced residues in the YSD1 channel do?

The ratchet mechanism has been proposed as a rotational movement induced by regularly spaced binding sites for the viral DNA on its egress path (Ref 13). This point has been removed from the DISCUSSION and is only mentioned p. 11, l. 295-299 as context for the structural

description of features that could facilitate DNA exit (p.12, l. 299-311). We focus our discussion on the description of the structural pattern that appears suitable for that purpose and conserved in T5 and possibly other *Siphoviridae*.

A comparison with the portal of phage phi29 is provided below. The spacing between successive rings of asp/glu differs from that of YSD1/T5; a couple of acidic residues are close in space and sequence (D194, E197) but have a different arrangement too. The fact that a negatively-charged interior surface is expected has been clarified (PMID 11130079, p12, l. 302).

12. *Here we show that to form the hexameric building blocks of the tail, the β -hairpin and N-terminal tail need to unmask complementary hydrophobic regions in the tail tube protein.*

Please see my earlier remarks about the actual evidence of the claimed property.

The discussion has been rewritten to integrate the new data and analysis. This statement is no longer in the discussion. In a related statement earlier in the text, we state that: “The location and amphipathic nature of the β -hairpin are compatible with a role in directly or indirectly shielding the hydrophobic surfaces involved inter-ring contacts (Fig. 5a and Supplementary Fig. 11b).”

- More than this, the precise arrangement of negative charges revealed here suggests a mechanistic explanation for the theoretical need for DNA to ratchet from the tube once the initial impetus to the DNA from the pressurized environment of the capsid begins to wane.

I am not sure that I follow the logics. The tube is negatively charged. It contains regularly spaced negatively charged amino acids as it should (it is a repetitive polymeric structure). All that repels the DNA and does not let it stick to the side of the tube. What ratchets are we discussing here?

We propose that the specific spacing of the repetitive Asp/Glu motifs provides preferred positioning of the DNA guiding it during egress. Experimental validation of the contribution of these motifs to DNA ejection efficiency is not currently possible in YSD1. However, the close match with DNA features (Fig. 6), sequence conservation (Fig. 5c,d) and structural

conservation in T5 (Fig. S9) provide convergent indication that these motifs have a role in the function of the tail. We have removed speculations about a specific mechanism from the DISCUSSION (cf. point 11 regarding ratcheting).

The text has been revised accordingly:

“The interior surface of the tube is lined with tracks of acidic residues along the length of the tail. These tracks present negatively-charged motifs with a longitudinal spacing that matches the theoretical sizes of the minor and major grooves of viral DNA. Apparent conservation of these motifs in phage T5 suggests they may have a role in guiding DNA during egress to facilitate its passage through the long and narrow conduit of the tail.

REVIEWERS' COMMENTS second round:

Reviewer #1 (Remarks to the Author):

The authors have addressed my concerns from the earlier version in an adequate manner.

Reviewer #2 (Remarks to the Author):

The revised MS "The architecture and stabilisation of flagellotropic tailed bacteriophages " is a much improved article. Thank you for addressing all referee's comments with such care (with figures, accurate citations, etc.). Very impressive and very much appreciated!

I have only two minor comments. Both are carryovers from the previous version.

1) The figure showing the crystal structure of the free capsomer subunit (Fig. 2b) is tiny. Smaller than a postmark it seems. This is rather unfortunate. This compact form of MCP has to be shown on a much larger scale. The differences between the free and shell-embedded conformations of the MCP must be discussed in greater detail. Would it be possible to discuss the energetics associated with this folding gymnastics? I believe that the fact that a structure of an HK97-fold protein is known in its free and capsid-integrated forms needs to be put front and center in this MS, as the chi MCP might become one of the most important proteins in the field of biophysics of virus shell assembly. The chi MCP is a goldmine for people studying protein folding.

2) Fig. 6a. How was the electrostatic change calculated? Which software used? How were the hydrogens added?

The architecture and stabilisation of flagellotropic tailed bacteriophages

Hardy, Dunstan et al.

We thank the reviewers for their insightful comments and suggestions that helped improve the manuscript. We provide a point-by-point response to the reviewers' points as follows.

REVIEWERS' COMMENTS:

Reviewer #1 (Remarks to the Author):

The authors have addressed my concerns from the earlier version in an adequate manner.

Reviewer #2 (Remarks to the Author):

The revised MS "The architecture and stabilisation of flagellotropic tailed bacteriophages" is a much improved article. Thank you for addressing all referee's comments with such care (with figures, accurate citations, etc.). Very impressive and very much appreciated!

I have only two minor comments. Both are carryovers from the previous version.

1) The figure showing the crystal structure of the free capsomer subunit (Fig. 2b) is tiny. Smaller than a postmark it seems. This is rather unfortunate. This compact form of MCP has to be shown on a much larger scale. The differences between the free and shell-embedded conformations of the MCP must be discussed in greater detail. Would it be possible to discuss the energetics associated with this folding gymnastics? I believe that the fact that a structure of an HK97-fold protein is known in its free and capsid-integrated forms needs to be put front and center in this MS, as the chi MCP might become one of the most important proteins in the field of biophysics of virus shell assembly. The chi MCP is a goldmine for people studying protein folding

We appreciate the reviewer's comments and have modified the figure accordingly by separating the old Figure 2 into Figure 2 and Figure 3. The differences in conformations are visible in greater details in the enlarged panels of Figure 2d,e,f.

The energetics of the folding gymnastics is a fascinating process but too complex to be addressed briefly in this manuscript. We trust that the transitions presented in this paper will warrant future studies to describe it in appropriate details from a theoretical and experimental point of view.

2) Fig. 6a. How was the electrostatic change calculated? Which software used? How were the hydrogens added?

The details were added to the "METHODS" section in the main text under "Model analysis".

"The surface potential and electrostatics were calculated with the adaptive Poisson-Boltzmann solver (APBS)⁶⁰ as implemented in PyMOL with hydrogen atoms added with the pdb2pqr method."